# Molecular Mechanisms of IL18 in Disease

**DOI:** 10.3390/ijms242417170

**Published:** 2023-12-06

**Authors:** Kyosuke Yamanishi, Masaki Hata, Naomi Gamachi, Yuko Watanabe, Chiaki Yamanishi, Haruki Okamura, Hisato Matsunaga

**Affiliations:** 1Department of Neuropsychiatry, Hyogo Medical University, 1-1 Mukogawa, Nishinomiya 663-8501, Hyogo, Japan; 2Department of Psychoimmunology, Hyogo Medical University, 1-1 Mukogawa, Nishinomiya 663-8501, Hyogo, Japan; 3Hirakata General Hospital for Developmental Disorders, Hirakata 573-0122, Osaka, Japan; wyuko@gen-info.osaka-u.ac.jp (Y.W.); hirochan@hirakataryoiku-med.or.jp (C.Y.)

**Keywords:** interleukin 18, inflammasome, diabetes, dyslipidemia, metabolic syndrome, brown adipose tissue, hippocampus, depression, learning and memory, Alzheimer’s disease

## Abstract

Interleukin 18 (IL18) was originally identified as an inflammation-induced cytokine that is secreted by immune cells. An increasing number of studies have focused on its non-immunological functions, with demonstrated functions for IL18 in energy homeostasis and neural stability. IL18 is reportedly required for lipid metabolism in the liver and brown adipose tissue. Furthermore, IL18 (*Il18*) deficiency in mice leads to mitochondrial dysfunction in hippocampal cells, resulting in depressive-like symptoms and cognitive impairment. Microarray analyses of *Il18*^−/−^ mice have revealed a set of genes with differential expression in liver, brown adipose tissue, and brain; however, the impact of IL18 deficiency in these tissues remains uncertain. In this review article, we discuss these genes, with a focus on their relationships with the phenotypic disease traits of *Il18*^−/−^ mice.

## 1. Introduction

Interleukin (IL) 18 was initially cloned in 1995 and identified as a proinflammatory cytokine that stimulates type 1 helper T cells to produce interferon (IFN)-γ [1]. The 23-kDa precursor form of IL18 is activated by cleaved caspase-1 and secreted as an active, 18-kDa mature form [2,3,4,5,6]. IL18 is secreted by hematopoietic lineages, such as macrophage cells [1] and microglia [7], as well as non-immune cells such as neural cells [6]. IL18 plays multiple roles in immune function, energy metabolism, and psychiatric disorders [1,8,9,10,11], and is also a therapeutic target for cancer immunotherapy, inhibition of body weight gain, and cognitive impairment [8,10,12]. Furthermore, we previously reported the effectiveness of the combination of IL18 and immune checkpoint inhibitors in suppressing tumor metastasis [12]. IL18 may exert anti-metastatic effects by increasing the numbers of effector-like natural killer (NK) cells or decreasing immunosuppressive cells, such as regulatory T cells. While IL18 alone can prime lymphocytes, IL18 combined with IL2 can promote the proliferation of NK cells, resulting in increased cytotoxicity against cancer [13]. Moreover, the expansion and function of NK cells stimulated by IL15 and IL18 are controlled by IL12 [14]. Together, this leads to a mechanism whereby antigen-mediated activation of dendritic cells leads to the production of cleaved caspase-1 and release of active IL18 and IL2, induced proliferation of NK cells, and secretion of IFN-γ. Thus, IL18 is an essential cytokine in inflammatory responses and cancer immunotherapy. 

Research has shown that IL18 is also closely associated with energy metabolism. In our studies on IL18-knockout (*Il18*^−/−^) mice, we observed a remarkable body weight gain over time in *Il18*^−/−^ mice compared with wild-type littermate mice [8,15]. Furthermore, *Il18*^−/−^ mice exhibited higher blood glucose and lipid levels, insulin resistance, non-alcoholic fatty liver disease (NAFLD), and non-alcoholic steatohepatitis (NASH) with aging. Regarding the mechanism of higher blood glucose levels and insulin resistance in *Il18*^−/−^ mice, the phosphorylation of signal transducer and activator of transcription 3 (STAT3) was impaired in the liver and was recovered by the administration of recombinant-IL18 (rIL18) [15]. Additionally, *Il18*^−/−^ mice with dyslipidemia, NAFLD, and NASH showed inhibition of the Wnt signaling pathway, reduced expression of cyclin D1 (*Ccnd1*), and disturbances in the circadian rhythm [8]. The dyslipidemia, NAFLD, and NASH in *Il18*^−/−^ mice were improved by administration of rIL18. We performed an additional study on brown adipose tissue (BAT) in these mice and confirmed that IL18 deficiency had similar effects to those observed in the liver [9]. Nevertheless, the direct molecular role of IL18 in dyslipidemia, NAFLD, and NASH remains to be clarified.

IL18 also plays a role in psychiatric and neurologic conditions [10,16]. We reported that *Il18*^−/−^ mice showed cognitive impairment and depressive-like behavioral changes compared with wild-type mice [10]. Another study on *Il18*^−/−^ mice revealed degenerated mitochondria in presynaptic axon terminals of the molecular and polymorphic layer of hippocampal dentate gyrus, suggesting the possible dysfunction of neurotransmitter release [10]. These disruptions may be related to the mechanism underlying the observed cognitive impairment and behavioral changes in *Il18*^−/−^ mice. Additionally, regulation of mitochondrial function may be involved in the role of IL18 in the brain. Furthermore, reports have identified several other genes, including transthyretin (*Ttr*), as having potential involvement in depression and cognitive impairment [10,17]. Therefore, the precise role and mechanisms of IL18 in brain dysfunction have not yet been clarified.

In this review, we summarize the current literature on genes showing differential expression (DEGs) between *Il18*^−/−^ and wild-type mice, focusing on those categorized as being related to energy metabolism, psychiatric conditions, and the brain, and discuss the potential relationships of these DEGs with IL18-deficient phenotypes in mice. Considering the role of IL18 in cancer, we also discuss the current literature and potential mechanisms of differentially expressed cancer-related genes. 

## 2. Microarray and Ingenuity Pathway Analysis (IPA) 

In our previous studies, we obtained microarray data from the liver, BAT, and brain of *Il18*^−/−^ mice at 12 weeks of age [8,9,10,17]. Genes with significantly increased or decreased expression compared with wild-type controls were extracted, and our analysis focused on DEGs that were commonly expressed in all three tissues (liver, BAT, and brain).

IPA was applied to analyze the functions of the DEGs, as previously described [18,19]. Of the tissue-specific DEGs, those associated with cancer, energy metabolism, and psychiatric and brain disorders (depression, dementia, Alzheimer’s disease [AD], and cognitive impairment) were selected on the basis of behaviors observed in *Il18*^−/−^ mice [10]. The IPA network explorer was run with default settings to reveal molecule–molecule interactions and to detect pathways between the molecules. 

The correlation between microarray and quantitative real-time PCR (RT-qPCR) results was analyzed using Spearman’s rank correlation tests. The rs and two-tailed *p* values for the microarray and RT-qPCR results of the liver, BAT, and brain were previously reported [8,9,17]. 

A diagram illustrating the interaction of the selected DEGs is provided in Figure 1.

## 3. IL18 and Cancer

An increasing number of studies have demonstrated a relationship between IL18 and cancer. In pancreatic ductal adenocarcinoma (PDA), serum and stromal IL18 is positively correlated with patient mortality [20,21]. High expression of IL18 in PDA was associated with worse disease progression and poor survival [22]. However, there is the other report that serum IL18 concentration was not correlated with patient survival of pancreatic adenocarcinoma [23]. In oral squamous cell carcinoma (OSCC), the serum levels of IL18 increase during tumor growth [24,25]. IL18 expression in peripheral blood mononuclear cells is also increased in OSCC patients compared with that in healthy individuals [25]. In OSCC patients with lymph node metastasis and a severe TNM stage, serum IL18 levels were significantly higher than those in patients without lymph node metastasis or a severe TNM stage. This trend has also been observed in patients with other cancers [25]. 

In clinical trials, systematic administration of IL18 significantly suppressed the growth of several kinds of carcinomas, such as melanoma and renal cell cancer, by stimulating the immune system [26,27,28]. Furthermore, we previously demonstrated the effectiveness of cancer immunotherapy using IL18 to augment immune checkpoint inhibitors [12]. Moreover, mutant IL18 engineered for resistance to inhibitory binding of the high-affinity IL-18 decoy receptor also promoted the activity of NK cells, resulting in the enhancement of anti-tumor effects in mouse tumor models [29]. These results suggest the possibility that IL18 may be an important cytokine in cancer treatment.

## 4. Cancer-Related Genes in *Il18^−/−^* Mice

Among the DEGs identified in our microarray analysis of *Il18*^−/−^ mice, those with involvement in various cancers are listed in Table 1. 

In human tongue squamous cell carcinoma cells, overexpression of IL18 led to apoptosis of tumor cells and decreased *Ccnd1* expression [30]. Though Jihong et al. reported that *Ccnd1* expression in the liver was unaffected by the administration of rIL18, they used BRL-3A rat liver cells to show that cell viability was increased with IL18 treatment [31]. In contrast, we found that IL18 administration increased the expression of *Ccnd1* in the liver of wild-type and *Il18*^−/−^ mice [8]. These results suggest that the IL18 receptor is expressed in the liver; however, the influence of IL18 on *Ccnd1* may occur through an indirect mechanism. 

*Erdr1* is related to cancer and IL18 [32,33]. Increased expression of *Erdr1* in mouse melanoma inhibited tumor growth and metastasis to the lung [32]. In another study, treatment with recombinant Erdr1 prevented invasion and migration of gastric cancer [34]. Overexpression of *Erdr1* has also been shown to suppress the expression of *Bcl2* and promote apoptosis [33,35,36]. Thus, *Erdr1* plays a role in cell homeostasis. One study has shown that Erdr1 is negatively regulated by IL18 in melanoma cells [32]. Erdr1 and Nfkb regulate the activation of STAT3, which increases tumor progression [37,38]. We observed that STAT3 phosphorylation was impaired in the liver of *Il18*^−/−^ mice, and was restored by r-IL18 treatment [15]. Erdr1 also functions as an immune activator that specifically activates NK cells [39]. Additionally, administration of recombinant ERDR1 augmented the cytotoxicity of primary human NK cells against leukemia cancer cells [39]. IL18 in combination with IL2 induces NK cells and has a cytotoxic role against cancers [13]. Consistent with other research [32], our microarray analysis showed that *Erdr1* expression was increased in liver, BAT, and brain of *Il18*^−/−^ mice, implying that *Erdr1* and *Il18* negatively regulate each other. We also reported decreased tumor sizes in lung metastasis models of melanoma treated with the combination of immune checkpoint inhibitors and IL18 [12]. Although *Erdr1*expression was not analyzed in our previous study, it is possible that the balance between *Il18* and *Erdr1* expression is important for the treatment of melanoma.

To the best of our knowledge, there are no previous reports on the relationship between *Nxpe4* and IL18. In our microarray analysis, the expression of *Nxpe4* was significantly decreased in all analyzed tissues. Colon cancer patients with decreased *Nxpe4* expression were found to have high mortality [40], and *Cygb*-deficient mice with increased colon tumors also exhibited decreased *Nxpe4* expression [41]. Therefore, future studies should examine whether *Il18*^-/-^ mice with decreased *Nxpe4* expression develop tumors over time.

Some reports have linked *Nnmt* and IL18 with cancer progression through the STAT3/IL1β pathway [42]. Our microarray analysis showed that the expression of *Nnmt* is significantly decreased in *Il18*^−/−^ mice. In PC-3 cells, a prostate cancer cell line, *Nnmt* promoted cell viability, whereas knockdown of *Nnmt* in PC-3 cells decreased cell viability [43]. Further studies are required to understand the functional relationship between IL18 and *Nnmt*.

No studies have examined the relationship between *Tmem25* and IL18. Only two studies have examined *Tmem25* expression in colon and breast cancer [44,45]. One clinical study showed that *Tmem25* expression was decreased in colorectal cancer; therefore, *Tmem25* has been considered as a therapeutic target [46]. Reduced *Tmem25* expression in breast cancer has been correlated with a better response to chemotherapy [44,45].

One study reported an association of *Atm* with IL18 [47]. In the colonic epithelium, the suppression of *Atm* reduces the activity of inflammasomes, including IL18, resulting in suppressed inflammation. In our microarray analysis of *Il18*^−/−^ mice, the expression of *Atm* was also reduced. Taken together, these results indicate a positive correlation between the expression of *Atm* and that of *Il18* [47]. *Atm* is reportedly a cancer risk factor [48,49]. One report showed that *Atm* variants and mutations are associated with the risk of pancreatic cancer [50,51]. Therefore, the relevance of the relationship between *Atm* and IL18 should be determined in further analyses. 

No studies have examined the relationship between IL18 and *Rps25*. One clinical research study revealed that *Rps25* expression was associated with the disease-specific survival rate in stage II mucinous colorectal cancer [52]. Furthermore, *Rps25* is considered a potential biomarker in lung adenocarcinoma and T-cell leukemia [53,54]. In *Il18*^−/−^ mice, the expression of *Rps25* is decreased. Thus, further study is required to clarify the interaction between these two factors.

The potential association between *Tmem267* with IL18 has also not been reported. One study showed a poor prognosis of liver and colon cancer in patients with elevated *Tmem267* levels [55]. Increased expression of *Tmem267* was also observed in *Il18*^−/−^ mice. Therefore, *Il18*^−/−^ mice may have a high risk of liver cancer.

The expression of *Axin2* in *Il18*^−/−^ mice was significantly decreased. *Axin2* is a critical modulator of the Wnt/β-catenin signaling pathway [56]. The *Axin2*-Wnt pathway involves negative feedback regulation, and *Axin2* is also a direct target of Wnt/β-catenin [56]. *Axin2* is a known tumor suppressor gene in some cancers [57]. In contrast, several reports identified *Axin2* as an oncogene in colorectal, liver, and gastric cancers [58]. *Axin2* is a β-catenin target that is highly expressed in human colorectal cancer [59]. Another study showed that the *Axin2* axis suppressed tumor growth and metastasis in colorectal cancer [60].

Caspase-4 (*Casp4*) is related to various malignancies and metastasis [61]. In non-small cell lung cancer (NSCLC) patients, the circulating *Casp4* level was much higher than that in healthy individuals. Furthermore, increased levels of *Casp4* in NSCLC patients led to higher mortality compared with those in NSCLC patients with low *Casp4* gene levels [62]. In gastric cancer patients, high expression of *Casp4* was associated with a better survival rate [63]. In esophageal squamous cell carcinoma, *Casp4* may be a tumor suppressor gene [64]. *Casp4* expression is decreased in *Il18*^−/−^ mice. Therefore, tumor growth might be increased in *Il18*^−/−^ mice compared with *Il18*^+/+^ mice.

Several reports have indicated the involvement of *Chrm1* in both the promotion and inhibition of cancer growth. *Chrm1* activates cholinergic signals and the hedgehog signaling pathway, resulting in the promotion of prostate cancer invasion [65,66]. Activation of *Chrm1* also induced the migration and invasion of two cancer cell lines, HepG2 and SMMC-7721, via the PI3K/Akt pathway [67]. Signaling through *Chrm1* inhibited primary pancreatic tumor growth via downregulation of the growth factor pathway [68]. In *Il18*^−/−^ mice, *Chrm1* expression was increased, indicating that tumor growth might be increased.

*Ifi16* reportedly functions as both a tumor suppressor and a promoter. High expression of *Ifi16* was observed in colorectal cancer [69,70]. Another study reported that *Ifi16* promoted cancer development in vitro and in vivo [71]. Ifi16 protein also activates the STING-TBK1 pathway for IFN-β production [72]. *Ifi16* functions as an activator of the inflammasome, resulting in the production of cleaved IL1β and IL18 [73]. One report showed that *Ifi16* suppresses cell viability and increases apoptosis in hepatocellular carcinoma (HCC) cell lines [74]. In *Il18*^−/−^ mice, the expression of *Ifi16* is significantly decreased. Further study is required to determine whether IL18 suppresses or promotes cancer development through *Ifi16*.

*Klf13* exhibits important functions in cell proliferation, migration, and differentiation [75,76]. *Klf13* inhibits cell proliferation and accelerates apoptosis in pancreatic cancer cells [77], and functions as a tumor suppressor protein in prostate cancer and colorectal cancer [78,79]. *Klf13* is also necessary for *Ccnd1* expression, which is an oncogene in oral squamous cell carcinoma [80]. *Klf13* and *Fgfr3* are highly expressed in oral cancer cells [81]. In *Il18*^−/−^ mice, the expression of *Klf13* is significantly increased. Therefore, tumor proliferation might be promoted in *Il18*^−/−^ mice.

Upregulated *Lrrc8e* led to cervical cancer cell proliferation and metastasis of breast cancer [82,83]. In *Il18*^−/−^ mice, the expression of *Lrrc8e* is decreased. *Lrrc8e* and *Il18* may be positively correlated; however, further study is warranted to determine whether IL18 can promote or suppress the growth of these tumor types.

In mouse models of cancer, LY6A has been identified as an important regulator of tumor progression [84,85,86]. LY6A exhibits marked influences on cellular activity and tumorigenicity, both in vitro and in vivo [87]. In *Il18*^−/−^ mice, the expression of *Ly6a* is decreased. Further study is needed to determine whether IL18 suppresses or promotes cancer progression through *Ly6a*.

*Nnt* is overexpressed in gastric cancer. *Nnt* accelerates tumor growth, lung metastasis, and peritoneal dispersion of cancer [88]. Furthermore, *Nnt* expression is upregulated in adrenocortical carcinoma and triggers anti-apoptosis pathways in cancer cells [89]. In mouse models of lung tumor initiation and progression, the expression of *Nnt* significantly enhances tumor growth, invasion, and aggressiveness [90]. Expression of *Nnt* is increased in *Il18*^−/−^ mice, indicating that tumor growth might be promoted.

Several previous studies have linked *Samsn1* and cancer. The human *SAMSN1* gene is located on chromosome 21q11-21, a region associated with heterozygous deletions frequently found in lung cancer cells, suggesting that *SAMSN1* may be a tumor suppressor [91,92]. Additionally, *SAMSN1* is a suppressive factor of multiple myeloma migration, both in vitro and in vivo [93]. Decreased expression of *SAMSN1* may promote the progression and recurrence of gastric cancer [94]. High expression of *Samsn1* was associated with high mortality in glioblastoma multiforme [95], but *Samsn1* was found to be expressed at significantly low levels in HCC [96]. In *Il18*^−/−^ mice, the expression of *Samsn1* is significantly decreased, suggesting that tumor growth might be promoted.

There are no reports on the involvement of *Npas1*, *Or10ad1*, *Ppcdc*, or *Wscd1* in cancer.

## 5. IL18 and Energy Metabolism

Previous studies have linked IL18 to energy metabolism, with potential roles in glucose and lipid homeostasis. High plasma levels of IL18 lead to a significant increase in the risk of type 2 diabetes (T2D), and serum levels of IL18 are significantly increased in patients with T2D compared with healthy controls [97,98,99]. Furthermore, IL18 levels in serum or plasma are negatively correlated with carbohydrate tolerance and positively related to insulin resistance [100,101,102]. High serum levels of IL18 increase the risk of metabolic syndrome characteristics such as hypertriglyceridemia, and are also linked to serum triglyceride levels [103,104]. In women with obesity, weight loss was found to reduce the levels of IL18 [105]. Plasma levels of IL18 were increased, but mRNA expression of *Il18* in adipose tissue was significantly decreased in obese mice compared with control mice [106]. Previous studies in *Il18*^−/−^ mice have indicated that IL18 is involved in glucose metabolism, lipid metabolism, and mitochondrial function [8,9,15]. IL18 deficiency was also shown to inhibit the phosphorylation of STAT3 in the liver, which may play a role in the mechanism of impaired energy metabolism, indicating the possible involvement of the Wnt signaling system [8]. 

## 6. Metabolism-Related Genes in *Il18^−/−^* Mice

DEGs identified in the microarray analysis of *Il18*^−/−^ mice that are involved in lipid and glucose metabolism are shown in Table 2.

*Atm* is required to maintain mitochondrial homeostasis [107]. Regulation of the DNA damage response by *Atm* involves inflammatory cytokines such as tumor necrosis factor-α and nuclear factor-κB [108]. Atm is implicated in intermediary metabolism through signaling pathways such as insulin and AMPK [109,110]. Aged *Atm*^−/−^ mice show an increase in blood glucose levels with lower insulin and C-peptide levels, whereas young *Atm*^−/−^ mice exhibit temporal hyperglycemia during oral glucose challenge comparing to age-matched wild-type controls [111,112]. *Atm*^−/−^ mice also display disturbances in carbohydrate metabolism, such as glucose intolerance, insulin resistance, and insufficient insulin secretion [111,113]. Furthermore, diet-induced hepatic steatosis is reduced in *Atm*^−/−^ mice compared with that in wild-type mice [114]. The *Atm* pathway is associated with oncogenesis [115]. In NASH, *Atm* mRNA expression accelerates signaling of oncogenic pathways [116,117]. Activation of *Atm* increases the accumulation of cholesterol [118]. In *Il18*^−/−^ mice, the expression of *Atm* is significantly decreased, raising the possibility that IL18 might regulate *Atm,* resulting in an imbalance of glucose and lipid metabolism.

*Casp4* is related to energy metabolism and responds to endoplasmic reticulum (ER) stress [119]. The ER is a crucial site of lipid metabolism, and a number of enzymes related to lipid metabolism exist there [120]. *Casp4* has been implicated in inflammasome activation through ER stress [121]. IL18 is an important component of the inflammasome. Decreased expression of *Casp4* was observed in *Il18*^−/−^ mice, which is consistent with these previous studies.

*Ccnd1* expression is decreased in *Il18*^−/−^ mice. We discussed the relationship between IL18, cyclin D1, and lipid metabolism in a previous study [8]. 

*Ifi16* is related to energy metabolism, and both lipid and glucose metabolism are affected by *Ifi16* expression [122]. One study showed that increased *Ifi16* expression stimulates adipogenesis in mice and humans [123]. Furthermore, the authors found that overexpression of *Ifi16* in mice led to obesity. Expression of *Ifi16* is significantly decreased in *Il18*^−/−^ mice, which leads us to speculate that *Ifi16* might not be related to obesity in *Il18*^−/−^ mice.

*Nnmt* is closely related to energy metabolism, and the expression of *Nnmt* in liver improves lipid parameters [124]. In humans and mice, the expression of *Nnmt* is negatively correlated with the levels of lipids, such as total cholesterol, low-density lipoprotein cholesterol, and triglycerides [125]. Another report showed that overexpression of *Nnmt* in mice led to fatty liver disease and fibrosis [126]. *Nnmt* expression in adipose tissue was also inversely correlated with insulin sensitivity [127]. In *Il18*^−/−^ mice, the expression of *Nnmt* is decreased. The phenotypes of *Il18*^−/−^ mice are partially consistent with some results of previous these papers.

No studies have examined the relationship between *Chrm1* and *Hmbs* expression and energy metabolism.

## 7. IL18 and Psychiatric Disorders

Previous studies have revealed that IL18 is closely related to several psychiatric disorders, including depressive disorders and schizophrenia [128]. Serum or plasma levels of IL18 in patients with depression, AD, or mild cognitive impairment were found to be higher than in healthy individuals [129,130]. Although IL18 is abnormally upregulated in neurons and glial cells in AD patients, IL18 levels are not associated with the severity of AD [131,132]. First-episode psychosis patients display increased plasma levels of IL18 that correlate with its severity [133]. An in vitro study using a human neuroblastoma cell line, Sh-sy5y, showed that IL18 promotes amyloid beta (Aβ) production and kinase activity, which is important for tau phosphorylation [134,135]. These findings indicate that IL18 is closely associated with various psychiatric disorders and cognitive impairment. Our study reported that IL18-deficient mice show depressive-like behavioral changes and impairments in learning and memory [10]. In another study, we reported that IL18 might have an adjustive function against stress [136]. 

## 8. Psychiatric and Brain Disorder-Related Genes in *Il18^−/−^* Mice

DEGs identified in the microarray analysis of *Il18*^−/−^ mice that are related to psychiatric disorders or psychiatric symptoms are shown in Table 3.

The *Atm* gene has multiple roles in central neurons. Previous studies have shown that *Atm* is required for apoptosis of the developing nervous system in response to DNA damage [137,138,139]. Deficiency of *Atm* in dentate gyrus led to decreased survival of proliferating neurons, suggesting that *Atm* may have a role in neural progenitor survival or differentiation in the hippocampus [140]. In *Il18*^−/−^ mice, the expression of *Atm* is significantly decreased, and neurogenesis in the hippocampus was suppressed in *Il18*^−/−^ mice compared with that in *Il18*^+/+^ mice [10]. Therefore, the histological phenotypes as suppressed neurogenesis in *Il18*^−/−^ mice may be caused by decreased expression of *Atm*.

*Casp4* is associated with the risk genes for AD [141]. *Casp4* expression is increased in the hippocampus and prefrontal cortex of CASP4/APP/PS1 mice, and increased expression of *Casp4* leads to hippocampal synaptic plasticity in APP/PS1 mice. *Casp4* is also expressed in microglia, and the presence of Casp4 led to more microglia clustered around amyloid plaques [141]. In *Il18*^−/−^ mice, the expression of *Casp4* is significantly decreased. In a previous study, less mature neural cells were observed in dentate gyrus of *Il18*^−/−^ mice (10); therefore, *Casp4* might be one of the genes responsible for the behavioral phenotypes of *Il18*^−/−^ mice.

*Chrm1* is related to various psychiatric disorders, including schizophrenia and mood disorders [142,143]. In patients with schizophrenia, the expression of *Chrm1* is decreased in the cortex [144]. *Chrm1* encodes a receptor that is highly expressed in glutamatergic neurons [145] and in postsynaptic regions of the hippocampus [146], and is a potential target molecule for schizophrenia treatment [142]. *Chrm1* expression is decreased in *Il18*^−/−^ mice, leading us to speculate that *Chrm1* may not be responsible for the behavioral phenotypes of *Il18*^−/−^ mice.

*Npas1* is expressed in neurons in the brain. Npas1 protein is a transcriptional suppressor of neuronal differentiation, development, and maturity functions [147,148]. Furthermore, *Npas1*-positive neural cells in the ventral pallidum modulate the susceptibility to stressors and anxiety-like behaviors [149]. In our previous study, *Il18*^−/−^ mice showed a depression-like phenotype, in which *Npas1* was upregulated and suppressed neurogenesis in the hippocampus [10]. While Npas1 may be one mediator of the depression-like phenotype in *Il18*^−/−^ mice [10], further analysis of the brain in these mice, particularly *Npas1*-positive neural cells, is warranted.

*Nnmt* is expressed in cholinergic neurons of the hippocampus [150]. In AD patients, *Nnmt* expression is increased in this area. In a post-mortem study, the expression of *Nnmt* in the prefrontal cortex was decreased in patients with schizophrenia compared with that in healthy individuals [151]. Patients with bipolar disorders also exhibit decreased serum levels of *Nnmt* compared with healthy individuals [152]. *Nnmt* expression is decreased in *Il18*^−/−^ mice. While *Nnmt* might be associated with several psychiatric disorders, further analysis of *Nnmt* in other brain regions, such as the hippocampus or prefrontal cortex, may improve our understanding of these relationships. 

## 9. Conclusions and Future Directions

In this review, we discussed the microarray analysis of DEGs identified in the liver, BAT, and brain of *Il18*^−/−^ mice. Building on a previous report linking IL18 deficiency to metabolic disorders and depressive-like behavioral changes, we describe here the relationship between IL18 and cancer tumorigenesis, and the potential for IL18 treatment of cancer immunotherapy in combination with immune checkpoint inhibitors. The DEGs commonly expressed in these three tissues may be related to cancer, energy metabolism, and psychiatric disorders. We speculate that the phenotypes of *Il18*^−/−^ mice may involve aberrations in these genes, and should be investigated in future studies. One limitation of this review is that it was based mainly on our own previous findings. As more research on *Il18* mutant mice is reported, thorough comparisons with our previous findings will be required. Although the many roles of IL18 have been revealed through basic and clinical studies (e.g., as a cancer therapy), there have been few clinical trials. Continued translational and clinical research is warranted to further investigate the potential of IL18 as a therapeutic agent.

## Figures and Tables

**Figure 1 ijms-24-17170-f001:**
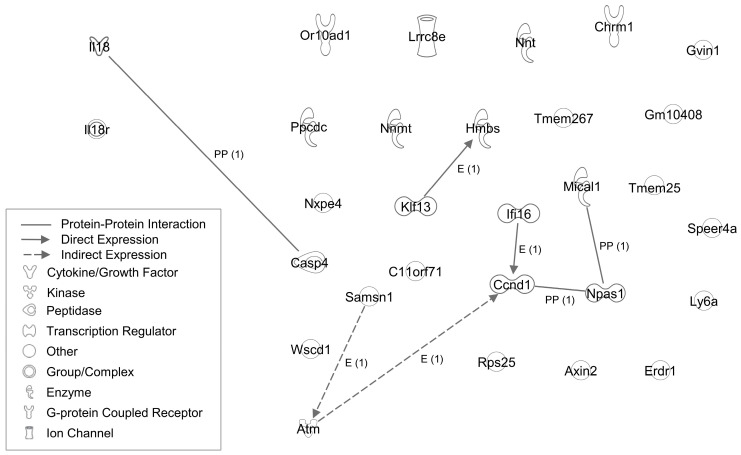
Ingenuity pathway analysis (IPA) diagram showing direct and indirect networks among interleukin (IL)18, its receptor (Il18r), and other differentially expressed genes common to liver, brown adipose tissue, and brain in *Il18*^−/−^ mice. Only *Casp4* was predicted to have a direct interaction with IL18.

**Table 1 ijms-24-17170-t001:** Cancers and related genes with differential expression in liver, brown adipose tissue, and brain under IL18 deficiency.

Symbol	Entrez Gene Name	Cancer
**Down-regulated genes**		
*Nxpe4*	neurexophilin and PC-esterase domain family member 4	colon cancer adenocarcinoma, carcinoma, melanoma
*Ifi16*	interferon gamma inducible protein 16	colorectal cancer, hepatocellular carcinoma
*Ccnd1*	cyclin D1	tongue squamous cell carcinoma
*Nnmt*	nicotinamide N-methyltransferase	prostate cancer
*Tmem25*	transmembrane protein 25	breast cancer, colon cancer
*Rps25*	ribosomal protein S25	colon rectal cancer, adenocarcinoma, T-cell leukemia
*Ppcdc*	phosphopantothenoylcysteine decarboxylase	endometrioid carcinoma, melanoma
*Axin2*	axin 2	colorectal cancer, liver cancer, gastric cancer
*Lrrc8e*	leucine rich repeat containing 8 VRAC subunit E	breast cancer
*Atm*	ATM serine/threonine kinase	pancreatic cancer
*Samsn1*	SAM domain, SH3 domain and nuclear localization signals 1	lung cancer, myeloma, gastric cancer, glioblastoma, HCC
*Ly6a*	lymphocyte antigen 6 complex, locus A	tumor progression
*Casp4*	caspase 4	lung cancer, gastric cancer, esophageal squamous cell carcinoma
**Up-regulated genes**		
*C11orf71*	chromosome 11 open reading frame 71	N.A.
*Wscd1*	WSC domain containing 1	N.A.
*Npas1*	neuronal PAS domain protein 1	alveolar rhabdomyosarcoma, soft tissue sarcoma cancer
*Hmbs*	hydroxymethylbilane synthase	N.A.
*Or10ad1*	olfactory receptor family 10 subfamily AD member 1	carcinoma, melanoma
*Klf13*	Kruppel like factor 13	oral cancer, pancreatic cancer, prostate cancer, colorectal cancer, oral squamous cell carcinoma
*Mical1*	microtubule associated monooxygenase, calponin and LIM domain containing 1	N.A.
*Tmem267*	transmembrane protein 267	liver cancer, colon caner
*Chrm1*	cholinergic receptor muscarinic 1	prostate cancer, pancreatic tumor
*Nnt*	nicotinamide nucleotide transhydrogenase	gastric cancer, adrenocortical carcinoma, lung tumor
*Erdr1*	erythroid differentiation regulator 1	melanoma, gastric cancers, leukemia cell cancer

N.A., not available.

**Table 2 ijms-24-17170-t002:** Glucose and lipid metabolism-related genes with differential expression in liver, brown adipose tissue, and brain under IL18 deficiency.

Symbol	Entrez Gene Name	Function
**Down-regulated genes**		
*Ifi16*	interferon gamma inducible protein 16	Glucose metabolism
*Ccnd1*	cyclin D1	Glucose metabolism, lipid metabolism
*Nnmt*	nicotinamide N-methyltransferase	Glucose metabolism
*Atm*	ATM serine/threonine kinase	Glucose metabolism, lipid metabolism
*Casp4*	caspase 4	Glucose metabolism
**Up-regulated genes**		
*Hmbs*	hydroxymethylbilane synthase	Glucose metabolism
*Chrm1*	cholinergic receptor muscarinic 1	Glucose metabolism, lipid metabolism

**Table 3 ijms-24-17170-t003:** Functions and diseases of psychiatric disorder-related genes with differential expression in liver, brown adipose tissue, and brain under IL18 deficiency.

Symbol	Entrez Gene Name	Function and Diseases
**Down-regulated genes**		
*Nnmt*	nicotinamide N-methyltransferase	Psychological disorders
*Atm*	ATM serine/threonine kinase	Learning, cognitive impairment
*Casp4*	caspase 4	Dementia, Alzheimer’s disease
**Up-regulated genes**		
*Npas1*	neuronal PAS domain protein 1	Learning, memory
*Chrm1*	cholinergic receptor muscarinic 1	Major depressive disorder, learning, memory, dementia, cognitive impairment, Alzheimer’s disease, psychological disorders

## Data Availability

The datasets shown and/or analyzed in the present study are available from the corresponding author upon reasonable request.

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
