# Peer review of "Molecular Mechanisms of IL18 in Disease"

_ijms, 2023, doi:10.3390/ijms242417170_

Round 1
Reviewer 1 Report
Comments and Suggestions for Authors
see word file

Comments on the Quality of English LanguageNA
Reviewer 2 Report
Comments and Suggestions for Authors
The inflammatory and metabolic conditions potentially associated to IL18 action
In this context, the review by presented here is a much-welcomed update.
One mild weakness of the manuscript resides in the fact that in some instances authors over rely on presentation of their own previous findings. For example, in line 70, the information “the series entries were GSE64309, GSE64308, and GSE64307, respectively” is somehow redundant, as certainly included in the cited references.
Line 73: as this is a review article, there is no need to identify the provider of the IPA program.
As presented, figure 1 must be re-edited. I believe that some elements of the IPA generated figure went missing.
Table 1 and 3 should be removed, as it presents primary data. Rather, the meaning and message of this gene expression changes should be critically discussed. The same applies to tables 4 and 5, that may be better represented as schematic figures. On the contrary, I think the summarizing nature of table 3 is appropriate for a review article.
Secondly, in lines 62-63 terminate the introductory part by alluding to some of their previous research. While this way of presentation would be appropriate for a primary research manuscript, I would suggest that authors decline to do so in this instance and rather they describe their previous accomplishments (with other relevant literature) in a dedicated chapter/paragraph.
Overall, the manuscript recapitulates in a very well-balanced manner the immunological and biological (metabolic and neurobiological) effects of IL-18.
This collection and discussion of the presented literature will be of intrinsic of interest to a wide audience, ranging from immunologists, basic scientists and clinicians alike.
Comments on the Quality of English Language
No majos problems, just double check for a few typing errors.
